# Effectiveness of an Active Offer of Influenza Vaccination to Hospitalized Frail Patients

**DOI:** 10.3390/vaccines13111165

**Published:** 2025-11-15

**Authors:** Alessandra Fallucca, Davide Anzà, Claudio Costantino, Cristina Genovese, Giovanni Genovese, Caterina Elisabetta Rizzo, Tania Vitello, Luigi Zagra, Vincenzo Restivo

**Affiliations:** 1Department of Health Promotion Sciences, Maternal and Infant Care, Internal Medicine and Excellence Specialties “G. D’Alessandro”, University of Palermo, Piazza delle Cliniche 2, 90127 Palermo, Italy; alessandra.fallucca@unipa.it (A.F.); claudio.costantino01@unipa.it (C.C.); tania.vitello@unipa.it (T.V.); luigi.zagra@unipa.it (L.Z.); 2School of Medicine, University Kore of Enna, Via delle Olimpiadi 4, 94100 Enna, Italy; davide.anza@unikore.it; 3Department of Biomedical Sciences and Morphological and Functional Images (BIOMORF), University of Messina, Via Consolare Valeria 1, 98124 Messina, Italy; crigenovese@unime.it (C.G.); gigenovese@unime.it (G.G.); caterinaelisabetta.rizzo@studenti.unime.it (C.E.R.)

**Keywords:** influenza vaccine, vaccine co-administration, health action process approach, older people, frail people, hospitalized people

## Abstract

**Background/Objectives**: Following the COVID-19 pandemic, the influenza season returned to its typical pre-pandemic circulation patterns. The category of people most vulnerable to severe influenza was older adults, and frail individuals, confirming their central role as a priority group for vaccination. The objective of this study was to evaluate the impact of an active influenza vaccination program in an area with low influenza vaccination rates and propensity to vaccine co-administration. **Methods**: People recruited were hospitalized frail individuals, patients over the age of 60, and those with chronic illnesses or comorbidities. It was administered a questionnaire to investigate adherence to influenza vaccination and the Health Action Process Approach was used to evaluate the propensity to co-administration. **Results**: A total of 418 hospitalized patients were enrolled in the study, of whom 58.4% (n = 244) received the influenza vaccine and 17.9% (n = 75) had a higher propensity to have co-administration of influenza and other recommended vaccines. The factors associated with influenza vaccination acceptance were received advice from hospital healthcare workers (aOR = 10.6 *p* < 0.001) and previous influenza vaccination (aOR = 18.1; *p* < 0.001). Propensity to vaccine co-administration was associated with a higher educational level (aOR = 4.21; *p* = 0.002), receiving vaccination advice from hospital healthcare workers (aOR = 2.80; *p* = 0.03), perceived positive outcome (aOR = 1.29; *p* = 0.02) and perceived self-efficacy (aOR = 1.48; *p* < 0.001). Conslusions: This study explored the impact on influenza vaccination coverage in implementing in hospital vaccination offer. The reliability of this strategy, together with the standard vaccination offer, could allow reaching the recommended vaccination coverage, particularly among at-risk people.

## 1. Introduction

According to the World Health Organization (WHO), seasonal influenza affects an estimated 1 billion people worldwide each year, with 3 to 5 million cases classified as severe and between 290,000 and 650,000 associated deaths [1].

During the 2022–2023 influenza season, which saw a return to typical pre-pandemic circulation patterns across the EU/EEA, the ECDC reported 5587 hospital admissions for laboratory-confirmed influenza, with a total of 367 associated deaths, of which 246 occurred in intensive care units and 121 in non-ICU settings [2].

Older adults, especially those aged 60 years and above, and frail individuals remained the most vulnerable to severe influenza, confirming their central role as a priority group for vaccination efforts [3]. Indeed, complications often arise from the exacerbation of pre-existing chronic conditions such as asthma, chronic obstructive pulmonary disease (COPD), diabetes, and cardiovascular diseases [4,5,6,7,8]. Vaccinating individuals with these conditions has proven effective in significantly reducing hospital admissions due to respiratory infections, influenza-related complications, and overall mortality—even among those not traditionally classified as high-risk [9]. In particular, vaccination has been shown to decrease the risk of hospitalization for heart and cardiovascular diseases by approximately 20% and to reduce the likelihood of stroke [10,11].

The 2023/24 Italian Ministry of Health guidelines recommend influenza vaccination for high-risk groups, including all adults aged 60 and over [12]. Moreover, in the 2023–2024 influenza vaccination campaign, the Sicilian Health Authority published a decree to vaccinate all categories at risk of complications by inviting all hospitals, nursing homes, and healthcare facilities to offer influenza vaccination to eligible patients before discharge [13]. In addition, the Sicilian Health Authorities strongly advised the co-administration of the influenza vaccine with other recommended vaccines—such as Pneumococcal, Herpes Zoster, Diphtheria-tetanus-pertussis (dTpa), and COVID-19 vaccines—whenever clinically appropriate [13]. Older people are prioritized not only because of low vaccine response due to immune senescence but also because approximately 75% of those aged 65–75 years and over 85% of those above 75 years have at least one chronic condition [12,14]. Although immunization strategies have historically prioritized childhood vaccination, recent public health policies have increasingly emphasized the importance of a life-course approach [15].

The target reported for the minimum vaccination coverage is 75% among older adults and individuals with chronic conditions [1,12]. However, in the 2023/24 season, only 53.3% of older adults in Italy received the influenza vaccine [16], which underscores that, despite strong recommendations for annual influenza vaccination as a key preventive tool, uptake in this group consistently falls below international targets.

As early as the 1990s, it was recognized that improving vaccination uptake among high-risk populations required comprehensive strategies, including educational initiatives, organizational enhancements, and targeted incentives for healthcare workers [17]. Multiple studies have shown that involving a wider range of healthcare professionals—beyond just primary care physicians—can significantly improve immunization rates in these groups [18]. Notably, older adults who are hospitalized during the influenza vaccination period often miss the opportunity to receive the vaccine, increasing their vulnerability to complications during the following influenza season [19]. Hospitals are a key opportunity to screen and immunize at-risk patients—such as those with chronic illnesses or older adults—during their stay and before discharge [20]. Active vaccination programs were widely implemented in many countries worldwide, but remain limited in Italy. A pilot study conducted in a Southern Italy hospital in 2022 showed a significant increase in influenza vaccination rates among the frail population through hospital-based interventions [21].

The objective of this study was to evaluate the impact of an active vaccination program in Sicily, where influenza vaccination rates among elderly and at-risk patients were notably low. The secondary outcome is to evaluate the associated factors of influenza vaccine acceptance. Finally, it was also explored the rate of propensity to vaccine co-administration and factors associated with higher propensity of vaccine co-administration.

## 2. Materials and Methods

### 2.1. Study Design and Context

Between November 2023 and February 2024, a multicentre study was conducted in Sicily. Hospitalized patients were recruited at Umberto I Hospital in Enna (156 beds), at the University Hospital of Messina (777 beds), and at the University Hospital of Palermo (526 beds), for an overall total of 1459 beds involved in the study.

### 2.2. Participants

The study population included frail individuals, patients over the age of 60, and those with chronic illnesses or comorbidities. Influenza vaccination was systematically offered to all eligible hospitalized patients (aged ≥ 60 years or with chronic conditions) before hospital discharge, independently of their level of frailty or personal attitudes. Information on previous influenza vaccination and receipt of in-hospital recommendations was collected through direct interviews and medical record verification, minimizing the risk of selection or recall bias.

### 2.3. Data Collection

Before data collection, the research team (public health physicians, resident physicians, and health assistants) trained the healthcare personnel on the objectives of the study and the methodologies employed. Collaboration with the medical and nursing staff and the consultation of the medical records allowed us to identify the hospital discharging patients. All patients were subsequently administered a validated, structured questionnaire. This study was approved by the Ethical Committee of Catania 2 at a meeting on 15 December 2023.

### 2.4. Measures

To investigate the factors associated with adherence to influenza vaccination among eligible hospitalized patients, a questionnaire was administered. The questionnaire was divided into three sections.

#### 2.4.1. Socio-Demographic Information

The first part explored socio-demographic information, including age, sex, weight, height, civil status, educational level, economic status, smoking habits, and vegetable and fruit consumption.

#### 2.4.2. Clinical Data

The second section gathered clinical data, including health status, comorbidities, previous vaccination against influenza, and vaccine advice from general practitioners.

#### 2.4.3. Health Action Process Approach (HAPA) Questionnaire

The third section assessed the propensity to co-administration of vaccines, utilizing validated constructs from the Health Action Process Approach (HAPA) questionnaire model, already used in studies conducted in the same administrative region [21]. The HAPA model explored four key domains relative to influenza vaccination: perceived risk, perceived positive outcome, perceived negative outcome, and perceived self-efficacy (Table 1). Participants’ responses were recorded using a five-point Likert scale, ranging from “strongly agree” to “strongly disagree.” Responses were subsequently converted to an ordinal scale, with 1 representing “strongly disagree” and 5 representing “strongly agree”.

**Table 1 vaccines-13-01165-t001:** Questions of the “Health Action Process Approach” about co-administration propensity.

Perceived risk	(a) Diseases such as influenza, COVID-19, pneumonia, or shingles could significantly worsen your health. (b) Influenza, COVID-19, pneumonia, or shingles could lead to a new hospitalization.
Perceived positive outcome	(c) Receiving two vaccines at the same time may reduce your risk of hospitalization due to disease-related complications. (d) Receiving two vaccines together may lower your risk of developing pneumonia during the colder months.
Perceived negative outcome	(e) Receiving two vaccines at the same time may reduce the frequency of local side effects, such as pain or redness at the injection site, compared to receiving a single vaccine. (f) Receiving two vaccines simultaneously may reduce the frequency of systemic side effects, such as fever or headache, compared to a single administration.
Perceived self-efficacy	(g) You feel confident that you have enough information about the simultaneous administration of multiple vaccines to make an informed decision. (h) You feel confident in your ability to decide to receive two vaccines at the same time, even if your family or friends disagree.

### 2.5. Statistical Analysis

The normality of the distributions for quantitative variables was assessed using the Skewness and Kurtosis test. Quantitative variables with a normal distribution were presented as mean (standard deviation), while those with a non-normal distribution were presented as median (interquartile range). For qualitative variables, the absolute and relative frequencies were calculated. The relationship between normally and non-normally distributed quantitative variables and higher propensity to have co-administration was analyzed using Student’s *t*-test or Wilcoxon and Mann–Whitney tests. Furthermore, the Chi-square test was used for qualitative variables. Finally, all variables that were significantly associated with a higher propensity to receive vaccine co-administration or influenza vaccine were included in a multivariate logistic regression model.

## 3. Results

### 3.1. Socio-Demographic Characteristics

A total of 418 hospitalized patients were enrolled in the study, of whom 58.4% (n = 244) received the influenza vaccine. Furthermore, 17.9% (n = 75) of participants had a higher propensity to have co-administration of influenza and other recommended vaccines. Approximately half of the participants were female (52.1%, n = 218), and 52% (n = 216) were aged over 65 years. The majority reported being married (60.3%, n = 252), having a low level of education (56.2%, n = 235), and a predominantly medium-to-high economic status (45.5%, n = 190). About lifestyle and dietary habits, 18.4% (n = 77) were obese, and nearly one-third (32%, n = 134) consumed at least three servings of fruits and vegetables per day. Only 23% (n = 96) were identified as current smokers (Table 2). Furthermore, statistically significant differences (Table 2) were observed between individuals who accepted influenza vaccination and those who did not. Vaccinated individuals were more likely to be aged 65 years or older (66.8% compared to 30.5%, *p* < 0.001). Marital status showed a weak but noteworthy association, with a slightly higher proportion of married individuals among the vaccinated group (63.1% vs. 56.3%, *p* = 0.02). Smoking habits also varied considerably, as current smokers were more prevalent among those who were not vaccinated (31.6% vs. 16.8%), whereas former smokers who had quit for over ten years were more common among those who received the vaccine (25.4% vs. 10.9%, *p* < 0.001). Additionally, individuals who were vaccinated reported higher daily consumption of fruits and vegetables, with 42.2% consuming three or more servings per day, compared to 30.4% among the unvaccinated (*p* = 0.05).

**Table 2 vaccines-13-01165-t002:** Socio-demographic information.

	Total (n = 418)	Accepted Administration of Flu Vaccine (n =244, 58.4%)	Declined Administration of Flu Vaccine *(n* = 174, 41.6%)	*p*	Higher Propensity of Vaccine Co-Administration *(n =* 75, 17.9%)	Lower Propensity of Vaccine Co-Administration *(n =* 343, 82.1%)	*p*
Gender							
Male	200 (47.9%)	121 (49.6%)	79 (45.4%)	0.4	34 (45.3%)	166 (48.4%)	0.6
Female	218 (52.1%)	123 (50.4%)	95 (54.6%)		41 (54.7%)	177 (51.6%)	
Age							
<65 years	202 (48.3%)	81 (33.2%)	121 (69.5%)	<0.001	26 (34.7%)	176 (51.3%)	0.009
≥65 years	216 (51.7%)	163 (66.8%)	53 (30.5%)		49 (65.3%)	167 (48.7%)	
BMI							
Underweight	7 (1.7%)	3 (1.2%)	4 (2.3%)	0.7	0 (0%)	7 (2%)	0.3
Normal weight	180 (43.1%)	103 (42.2%)	77 (44.2%)		36 (48%)	144 (42%)	
Overweight	154 (36.8%)	90 (36.9%)	64 (36.8%)		27 (36%)	127 (37%)	
Type I Obesity	48 (11.5%)	28 (11.5%)	20 (11.5%)		10 (13.3%)	38 (11.1%)	
Type II Obesity	29 (6.9%)	20 (8.2%)	9 (5.2%)		2 (2.7%)	27 (7.9%)	
Civil status							
Married	252 (60.3%)	154 (63.1%)	98 (56.3%)	0.02	43 (57.3%)	209 (60.9%)	0.9
Engaged	11 (2.6%)	5 (2%)	6 (3.5%)		3 (4%)	8 (2.3%)	
Cohabiting partner	26 (6.2%)	11 (4.5%)	15 (8.6%)		4 (5.3%)	22 (6.4%)	
Divorced	22 (5.3%)	10 (4.1%)	12 (6.9%)		5 (6.7%)	17 (5%)	
Single	47 (11.2%)	21 (8.6%)	26 (14.9%)		8 (10.7%)	39 (11.4%)	
Widowed person	60 (14.4%)	43 (17.6%)	17 (9.8%)		12 (16%)	48 (14%)	
Educational level							
Low	235 (56.2%)	145 (59.4%)	90 (51.7%)	0.25	33 (44%)	202 (58.9%)	0.002
Medium	127 (30.4%)	67 (27.5%)	60 (34.5%)		23 (30.7%)	104 (30.3%)	
High	56 (13.4%)	32 (13.1%)	24 (13.8%)		19 (25.3%)	37 (10.8%)	
Economic status							
Low	41 (9.8%)	24 (9.8%)	17 (9.8%)	0.99	7 (9.3%)	34 (9.9%)	0.6
Low-Medium	166 (39.7%)	96 (39.3%)	70 (40.2%)		25 (33.3%)	141 (41.1%)	
Medium-High	190 (45.5%)	112 (45.9%)	78 (44.8%)		38 (50.7%)	152 (44.3%)	
High	21 (5%)	12 (4.9%)	9 (5.2%)		5 (6.7%)	16 (4.7%)	
Smoking status							
No, never smoked	279 (66.7%)	182 (74.6%)	97 (55.7%)	<0.001	57 (76%)	222 (64.7%)	0.17
Active smoker	96 (23%)	41 (16.8%)	55 (31.6%)		13 (17.3%)	83 (24.2%)	
Quit smoking < 10 years ago	43 (10.3%)	21 (8.6%)	22 (12.7%)		5 (6.7%)	38 (11.1%)	
Fruit and vegetable intake (servings/day)							
none	38 (9.1%)	17 (7%)	21 (12.1%)	0.05	4 (5.3%)	34 (9.9%)	0.2
1 or 2 servings	224 (53.6%)	124 (50.8%)	100 (57.5%)		36 (48%)	188 (54.8%)	
3 or 4 servings	134 (32%)	87 (35.7%)	47 (27%)		29 (38.7%)	105 (30.6%)	
5 or more servings	22 (5.3%)	16 (6.5%)	6 (3.4%)		6 (8%)	16 (4.7%)	

BMI = Body mass index; SD = Standard deviation.

Among those who accepted co-administration of vaccines, significant differences were also evident. Individuals aged 65 years and older were more frequently represented in the co-administration group compared to those who declined it (65.3% vs. 48.7%, *p* = 0.009). Educational level was another relevant factor, as a higher proportion of those accepting co-administration had attained a high level of education (25.3% vs. 10.8%), while a low level of education was more common among those who refused it (58.9% vs. 44%, *p* = 0.002).

### 3.2. Influenza Vaccine Uptake and Associated Factors

About the health status of the interviewed patients (Table 3), the majority described their condition as fair (42.3%, n = 177). More than one-fifth (21.5%, n = 90) reported having comorbidities, the most common of which were hypertension (38.3%, n = 160), other cardiovascular diseases (26.3%, n = 110), and respiratory diseases (21.5%, n = 90). Additionally, 47.6% (n = 199) reported suffering from two or more chronic conditions.

In terms of influenza vaccination, slightly more than half of the participants (54.1%, n = 226) stated that they had received a recommendation to vaccinate from their general practitioner, while 56% (n = 234) reported receiving the same recommendation from hospital healthcare personnel. Overall, 53.3% (n = 223) had received an influenza vaccination at least once in the past; however, only 38% (n = 159) were vaccinated during the 2022/2023 influenza season. By contrast, most respondents (92.6%, n = 387) reported having been vaccinated against SARS-CoV-2 in the past.

Statistically significant differences (Table 3) were observed between participants who accepted influenza vaccination and those who did not. Individuals in the vaccinated group were more likely to present with comorbidities such as diabetes (17.2% vs. 8.6%; *p* = 0.01) and hypertension (47.5% vs. 25.3%; *p* < 0.001). They were also significantly more likely to have received medical advice recommending vaccination from hospital staff (76.2% vs. 27.6%; *p* < 0.001), to have a history of influenza vaccination (78.3% vs. 18.4%; *p* < 0.001), and to be vaccinated during the current season (62.7% vs. 3.4%; *p* < 0.001). Prior COVID-19 vaccination was also more common among those who received the influenza vaccine (97.2% vs. 86.2%, *p* < 0.001).

Among individuals who accepted co-administration of vaccines, these trends were even more pronounced. Higher rates of diabetes (25.3% vs. 11.1%) and hypertension (56.0% vs. 34.4%) were observed compared to those who declined co-administration. In this group, prior COVID-19 vaccination was also significantly more frequent (97.3% vs. 91.5%, *p* = 0.01).

**Table 3 vaccines-13-01165-t003:** Health status, comorbidities, and vaccination history by influenza vaccine and co-administration acceptance.

	Total (n = 418)	Accepted Administration of Influenza Vaccine (n =244, 58.4%)	Declined Administration of Influenza Vaccine *(n =* 174, 41.6%)	*p*	Accepted Co-Administration of Influenza Vaccine *(n =* 75, 17.9%)	Declined Co-Administration of Influenza Vaccine *(n =* 343, 82.1%)	*p*
Health status							
Very poor	12 (2.9%)	7 (2.9%)	5 (2.9%)	0.02	1 (1.3%)	11 (3.2%)	0.16
Poor	88 (21.1%)	59 (24.2%)	29 (16.7%)		13 (17.3%)	75 (21.9%)	
Fair	177 (42.3%)	111 (45.5%)	66 (37.9%)		39 (52%)	138 (40.2%)	
Good	108 (25.8%)	54 (22.1%)	54 (31%)		20 (26.7%)	88 (25.7%)	
Very good	33 (7.9%)	13 (5.3%)	20 (11.5%)		2 (2.7%)	31 (9%)	
Comorbidities							
Respiratory disease							
No	328 (78.5%)	182 (74.6%)	146 (83.9%)	0.02	62 (82.7%)	266 (77.5%)	0.3
Yes	90 (21.5%)	62 (25.4%)	28 (16.1%)		13 (17.3%)	77 (22.5%)	
Cardiovascular disease							
No	308 (73.7%)	169 (69.3%)	139 (79.9%)	0.02	55 (73.3%)	253 (73.8%)	0.9
Yes	110 (26.3%)	75 (30.7%)	35 (20.1%)		20 (26.7%)	90 (26.2%)	
Diabetes							
No	361 (86.4%)	202 (82.8%)	159 (91.4%)	0.01	56 (74.7%)	305 (88.9%)	0.001
Yes	57 (13.6%)	42 (17.2%)	15 (8.6%)		19 (25.3%)	38 (11.1%)	
Kidney failure							
No	381 (91.2%)	217 (88.9%)	164 (94.2%)	0.06	67 (89.3%)	314 (91.5%)	0.5
Yes	37 (8.8%)	27 (11.1%)	10 (5.8%)		8 (10.7%)	29 (8.5%)	
Stroke							
No	383 (91.6%)	222 (91%)	161 (92.5%)	0.57	70 (93.3%)	313 (91.2%)	0.6
Yes	35 (8.4%)	22 (9%)	13 (7.5%)		5 (6.7%)	30 (8.8%)	
Cancer							
No	387 (92.6%)	223 (91.4%)	164 (94.2%)	0.27	69 (92%)	318 (92.7%)	0.8
Yes	31 (7.4%)	21 (8.6%)	10 (5.8%)		6 (8%)	25 (7.3%)	
Liver disease							
No	366 (87.6%)	210 (86.1%)	156 (89.7%)	0.27	71 (94.7%)	295 (86%)	0.04
Yes	52 (12.4%)	34 (13.9%)	18 (10.3%)		4 (5.3%)	48 (14%)	
Hypertension							
No	258 (61.7%)	128 (52.5%)	130 (74.7%)	<0.001	33 (44%)	225 (65.6%)	<0.001
Yes	160 (38.3%)	116 (47.5%)	44 (25.3%)		42 (56%)	118 (34.4%)	
Rheumatologic disease							
No	386 (92.3%)	231 (94.7%)	155 (89.1%)	0.03	74 (98.7%)	312 (91%)	0.02
Yes	32 (7.7%)	13 (5.3%)	19 (10.9%)		1 (1.3%)	31 (9%)	
Comorbidity							
No	219 (52.4%)	106 (43.4%)	113 (64.9%)	<0.001	37 (49.3%)	182 (53.1%)	0.4
2 diseases	104 (24.9%)	70 (28.7%)	34 (19.6%)		23 (30.7%)	81 (23.6%)	
>2 diseases	95 (22.7%)	68 (27.9)	27 (15.5%)		15 (20%)	80 (23.3%)	
Hospital ward							
Clinical	231 (55.3%)	154 (63.1%)	77 (44.2%)	<0.001	-	-	-
Surgical	187 (44.7%)	90 (36.9%)	97 (55.8%)		-	-	
Vaccine advice from General Practitioner							
No	192 (45.9%)	70 (28.7%)	122 (70.1%)	<0.001	30 (40%)	162 (47.2%)	0.3
Yes	226 (54.1%)	174 (71.3%)	52 (29.9%)		45 (60%)	181 (52.8%)	
Vaccine advice from Hospital Healthcare Workers							
No	184 (44%)	58 (23.8%)	126 (72.4%)	<0.001	11 (14.7%)	173 (50.4%)	<0.001
Yes	234 (56%)	186 (76.2%)	48 (27.6%)		64 (85.3%)	170 (49.6%)	
Vaccinated for Influenza previously							
No	167 (40%)	37 (15.2%)	130 (74.7%)	<0.001	13 (17.3%)	154 (44.9%)	<0.001
Yes	223 (53.3%)	191 (78.3%)	32 (18.4%)		53 (70.7%)	170 (49.6%)	
Not sure	28 (6.7%)	16 (6.6%)	12 (6.9%)		9 (12%)	19 (5.5%)	
Vaccinated for Influenza during the 2022–2023 season							
No	259 (62%)	91 (37.3%)	168 (96.6%)	<0.001	27 (36%)	232 (67.6%)	<0.001
Yes	159 (38%)	153 (62.7%)	6 (3.4%)		48 (64%)	111 (32.4%)	
Vaccinated for COVID-19 previously							
No	27 (6.4%)	4 (1.6%)	23 (13.2%)	<0.001	0 (0%)	27 (7.9%)	0.01
Yes	387 (92.6)	237 (97.2%)	150 (86.2%)		73 (97.3%)	314 (91.5%)	
Not sure	4 (1%)	3 (1.2%)	1 (0.6)		2 (2.7%)	2 (0.6%)	

### 3.3. Vaccine Co-Administration Propensity and Group Differences

The analysis of HAPA-related psychological constructs revealed significant differences between patients who expected to accept co-administration of the influenza vaccine and those who did not (Table 4). Specifically, the median “perceived risk” score was 8 (IQR: 7–10) among those who had more propensity to co-administration, compared to 6 (IQR: 4–8) among those who did not (*p* < 0.001), with a greater proportion of individuals scoring above the median in the more propensity group (81.3% vs. 44.6%, *p* < 0.001). Similarly, “perceived positive outcome” had a median score of 8 (IQR: 8–10) among those who had more propensity to co-administration versus 6 (IQR: 4–8) among those who declined (*p* < 0.001), with 96.0% of the more propensity group scoring above the median compared to 64.4% who did not (*p* < 0.001). For “perceived negative outcome,” the median score was 8 (IQR: 8–8) among who had more propensity to co-administration and 6 (IQR: 6–8) among who did not (*p* < 0.001), with a higher proportion of more propensity group again scoring above the median (82.7% vs. 45.8%, *p* < 0.001). Finally, “perceived self-efficacy” had the lowest overall scores, but still showed a significant difference: a median of 8 (IQR: 6–8) among those who had more propensity to co-administration versus 6 (IQR: 4–6) among those who did not (*p* < 0.001). In this case, 89.3% of the more propensity group scored above the median, compared to only 53.9% who did not.

**Table 4 vaccines-13-01165-t004:** HAPA mode of vaccine co-administration.

	Total (n = 418)	Accepted Vaccine Co-Administration *(n =* 75, 17.9%)	Declined Vaccine Co-Administration *(n =* 343, 82.1%)	*p*
Perceived risk				
Median score (IQR)	7 (6–8)	8 (7–10)	6 (4–8)	<0.001
Low (<median)	204 (48.8%)	14 (18.7%)	190 (55.4%)	<0.001
High (>median)	214 (51.2%)	61 (81.3%)	153 (44.6%)	
Perceived positive outcome				
Median score (IQR)	6 (4–8)	8 (8–10)	6 (4–8)	<0.001
Low (<median)	125 (29.9%)	3 (4.0%)	122 (35.6%)	<0.001
High (>median)	293 (70.1%)	72 (96.0%)	221 (64.4%)	
Perceived negative outcome				
Median score (IQR)	7 (6–8)	8 (8–8)	6 (6–8)	<0.001
Low (<median)	199 (47.6%)	13 (17.3%)	186 (54.2%)	<0.001
High (>median)	219 (52.4%)	62 (82.7%)	157 (45.8%)	
Perceived self-efficacy				
Median score (IQR)	6 (4–8)	8 (6–8)	6 (4–6)	<0.001
Low (<median)	166 (39.7%)	8 (10.7%)	158 (46.1%)	<0.001
High (>median)	252 (60.3%)	67 (89.3%)	185 (53.9%)	

IQR = interquartile range.

### 3.4. Multivariable Analysis

The multivariable logistic regression model for influenza vaccination acceptance showed that having received advice from hospital healthcare workers (aOR = 10.6; 95% CI: 5.33–20.9; *p* < 0.001) and previous influenza vaccination (aOR = 18.1; 95% CI: 7.98–41.1; *p* < 0.001) were both significantly associated with current vaccine uptake.

Furthermore, regarding the propensity to vaccine co-administration, a higher educational level was positively associated with co-administration (aOR = 4.21; 95% CI: 1.69–10.49; *p* = 0.002), as was receiving vaccination advice from hospital healthcare workers (aOR = 2.80; 95% CI: 1.13–6.93; *p* = 0.03). Moreover, psychological factors such as perceived positive outcome (aOR = 1.29; 95% CI: 1.05–1.59; *p* = 0.02) and perceived self-efficacy (aOR = 1.48; 95% CI: 1.22–1.8; *p* < 0.001) were significantly associated with the likelihood of more propensity to co-administration (Table 5).

**Table 5 vaccines-13-01165-t005:** Univariable and multivariable analysis of influenza vaccine uptake and co-administration propensity.

	Influenza Vaccine Administration Uptake						Vaccines Co-Administration Propensity					
	Univariable Analysis			Multivariable Analysis			Univariable Analysis			Multivariable Analysis		
	cOR	CI 95%	*p*	aOR	CI 95%	*p*	cOR	CI 95%	*p*	aOR	CI 95%	*p*
Male vs. female	0.9	0.57–1.25	0.4	0.64	0.34–1.18	0.15	1.13	0.68–1.87	0.63	1.24	0.67–2.30	0.49
Age <65 vs. ≥65 years	4.6	3.02–7	<0.001	1.74	0.82–3.66	0.15	1.98	1.18–3.34	0.01	1.31	0.62–2.79	0.48
BMI												
Normal vs. underweight	1,78	0.39–8.2	0.5	-	-	-	3.37	0.77–14.86	0.1	-	-	-
Overweight vs. normal weight	1.9	0.4–8.7	0.4	-	-	-	2.87	0.64–12.8	0.17	-	-	-
Obese I vs. normal weight	1.87	0.38–9.3	0.45	-	-	-	3.55	0.72–17.53	0.12	-	-	-
Obese II vs. normal weight	2.97	0.55–16.08	0.2	-	-	-	-	-	-	-	-	-
Civil status				-	-	-						
Engaged	0.53	1.16–1.78	0.3	-	-	-	1.82	0.46–7.1	0.39	-	-	-
Cohabiting partner	0.47	0.2–1.05	0.07	-	-	-	0.88	0.29–2.7	0.82	-	-	-
Divorced	0.53	0.22–1.27	0.15	-	-	-	1.42	0.5–4.08	0.5	-	-	-
Single	0.51	0.27–0.96	0.04	-	-	-	1	0.43–2.28	0.99	-	-	-
Widowed person	1.61	0.87–2.99	0.13	-	-	-	1.22	0.59–2.48	0.59	-	-	-
Economic status				-	-	-						
High	0.97	0.49–1.94	0.93	-	-	-	1.52	0.42–5.53	0.53	-	-	-
Medium-High	1.01	0.51–2.02	0.96	-	-	-	1.21	0.5–2.95	0.67	-	-	-
Medium-Low	0.94	0.33–2.74	0.92	-	-	-	0.86	0.34–2.16	0.75	-	-	-
Educational level				-	-	-						
Medium	0.69	0.45–1.07	0.1	-	-	-	1.35	0.76–2.42	0.3	2.10	0.99–4.46	0.05
High	0.83	0.46–1.49	0.53	-	-	-	3.14	1.62–6.1	0.001	4.21	1.69–10.49	0.002
Smoking status												
Active smoker	0.48	0.3–0.79	0.004	1.1	0.5–2.42	0.82	0.68	0.34–1.35	0.27	-	-	-
Quit smoking < 10 years	0.62	0.32–1.2	0.16	0.71	0.26–1.97	0.51	0.57	0.21–1.55	0.27	-	-	-
Quit smoking > 10 anni	2.12	1.18–3.81	0.01	1.45	0.59–3.51	0.42	1.43	0.77–2.64	0.26	-	-	-
Fruit and vegetable intake (servings/day)												
1 or 2 servings	1.53	0.77–3.06	0.23	0.9	0.28–2.89	0.86	1,63	0.54–4.87	0.38	-	-	-
3 or 4 servings	2.29	1.1–4.75	0.03	1.6	0.48–5.37	0.44	2.35	0.77–7.16	0.13	-	-	-
5 or more servings	3.29	1.06–10.25	0.04	1.88	0.35–10.27	0.46	3.19	0.79–12.9	0.1	-	-	-
Health status												
Poor	1.45	0.42–4.98	0.55	-	-	-	1.9	0.23–16.05	0.55	-	-	-
Fair	1.2	0.37–3.93	0.76	-	-	-	3.1	0.39–24.83	0.28	-	-	-
Good	0.71	0.21–2.4	0.59	-	-	-	2.5	0.3–20.5	0.39	-	-	-
Very good	0.46	0.12–1.78	0.26	-	-	-	0.71	0.58–8.62	0.79	-	-	-
Comorbidities												
Respiratory disease	1.78	1.08–2.92	0.02	1.43	0.61–3.38	0.41	0.72	0.38–1.39	0.33	-	-	-
Cardiovascular disease	1.76	1.11–2.79	0.02	0.82	0.33–2.03	0.66	1.02	0.58–1.8	0.93	-	-	-
Diabetes	2.2	1.18–4.12	0.01	1.3	0.48–3.49	0.6	2.73	1.46–5.06	0.002	2.00	0.91–4.44	0.08
Kidney failure	2.04	0.96–4.33	0.06	-	-	-	1.29	0.57–2.95	0.54	-	-	-
Stroke	1.23	0.6–2.5	0.58	-	-	-	0.74	0.28–1.99	0.56	-	-	-
Cancer	1.54	0.7–3.37	0.28	-	-	-	1.1	0.44–2.8	0.83	-	-	-
Liver disease	1.4	0.76–2.58	0.28	-	-	-	0.35	0.12–0.99	0.05	0.86	0.23–3.26	0.82
Hypertension	2.68	1.75–4.09	<0.001	2.04	0.96–4.34	0.06	2.42	1.46–4.03	0.001	1.61	0.80–3.24	0.18
Rheumatologic disease	0.46	0.22–0.96	0.04	0.52	0.17–1.6	0.26	0.14	0.02–1.01	0.05	-	-	-
Comorbidity												
2 diseases	2.19	1.35–3.58	0.002	0.83	0.33–2.09	0.69	1.4	0.78–2.5	0.26			
>2 diseases	2.68	1.6–4.5	<0.001	0.52	0.16–1.69	0.28	0.92	0.48–1.77	0.8			
Hospital ward												
Clinical vs. Surgical	2.2	1.45–3.2	<0.001	1.37	0.72–2.6	0.34	-	-	-	-	-	-
Vaccine advice from General Practitioner												
Yes vs. no	5.83	3.8–8.9	<0.001	1.51	0.76–3	0.24	1.34	0.81–2.23	0.26	-	-	-
Vaccine advice from Hospital Healthcare Workers												
Yes vs. no	8.41	5.4–13.13	<0.001	10.6	5.33–20.9	<0.001	5.9	3.02–11.6	<0.001	2.80	1.13–6.93	0.03
Vaccinated for Influenza previously												
Yes vs. no	20.97	12.4–35.4	<0.001	18.1	7.98–41.1	<0.001	3.69	1.94–7.04	<0.001	1.03	0.3–3.52	0.18
Not sure vs. no	4.68	2.04–10.8	<0.001	1.89	0.64–5.57	0.25	5.61	2.12–14.87	0.001	1.89	0.55–6.46	0.31
Vaccinated for Influenza during 2022–2023 season												
Yes vs. no	47.08	20.03–110.66	<0.001	-	-	-	3.71	2.2–6.27	<0.001	1.70	0.59–4.91	0.33
Vaccinated for COVID-19 previously												
Yes vs. no	9.08	3.08–26.8	<0.001	4.18	0.94–18.67	0.06	0.23	0.32–1.68	0.15	-	-	-
Not sure vs. no	17.25	1.42–210.12	0.03	3.43	0.15–77.55	0.44	-	-	-	-	-	-
Perceived risk	-	-	-	-	-	-	1.47	1.28–1.69	<0.001	1.00	0.80–1.25	0.99
Perceived positive outcome	-	-	-	-	-	-	1.73	1.48–2.03	<0.001	1.29	1.05–1.59	0.02
Perceived negative outcome	-	-	-	-	-	-	1.41	1.22–1.63	<0.001	1.10	0.90–1.35	0.34
Perceived self-efficacy	-	-	-	-	-	-	1.74	1.49–2.04	<0.001	1.48	1.22–1.80	<0.001

## 4. Discussion

This study investigated the determinants of influenza vaccine uptake among hospitalized patients during the 2023/2024 season in three Sicilian hospitals, with a particular focus on the impact of proactive in-hospital vaccination counselling, as well as the behavioural factors influencing participants’ attitudes toward the vaccine co-administration. Notably, only 17.9% of enrolled patients had more propensity to co-administration, underscoring ongoing hesitancy or uncertainty toward receiving two vaccines simultaneously.

A distinctive feature of this survey was the active offer of influenza vaccination by medical researchers directly within hospital wards. This proactive, provider-driven approach involved face-to-face interactions, where healthcare professionals not only provided information but also strongly recommended and offered vaccination during a period of heightened perceived vulnerability. According to the literature, personalized vaccine recommendations—particularly when delivered by trusted healthcare professionals—are among the most effective predictors of vaccine uptake [21,22]. Unlike passive informational strategies such as reminder/recall systems (e.g., SMS, emails, letters), mass media campaigns (e.g., posters, videos), digital tools (e.g., apps, web portals, chatbots), or automated provider alerts, this approach relies on direct interpersonal communication initiated by healthcare providers at the patient’s bedside. It capitalizes on hospitalization as a key opportunity to engage patients in preventive health behaviours and may be especially beneficial for individuals with low adherence to outpatient care, by integrating vaccination into routine inpatient management [15,20]. However, this model remains underutilized in the Italian healthcare setting, despite previous evidence supporting its feasibility and effectiveness [20,21,22,23,24].

Notably, *Fallucca et al*. conducted a pioneering survey during the 2022/2023 influenza season at the Palermo University Hospital, reporting a vaccination uptake of 62.5% [21]. Compared to the previous investigation, our multicentric study observed a slightly lower uptake rate of 58.4%. This slight decline may be explained by the fact that in some participating centres, this was the first implementation of a similar preventive strategy and, as often happens when introducing a new public health intervention, it is essential to ensure standardization of procedures and communication between healthcare workers. Nevertheless, this result remains encouraging when considered alongside national and regional influenza vaccination coverage among older adults. In fact, during the same influenza season, coverage among individuals in the previous season was 20% lower, suggesting that hospital-based vaccination strategies—particularly if extended to other hospital departments and healthcare facilities—may achieve higher coverage rates than standard community-based strategies both at the regional and national level.

Multivariate analysis confirmed the pivotal role of hospital-based interventions. Receiving a vaccination recommendation from hospital staff was associated with a more than tenfold increase in the likelihood of influenza vaccine uptake (aOR = 10.6; 95% CI: 5.33–20.9; *p* < 0.001), consistent with findings from other investigations on inpatient vaccination adherence [20,22]. Similarly, counseling by healthcare professionals was associated with more than a threefold increase in the likelihood of accepting co-administration of both vaccines (*p* = 0.004). This underscores the strategic value of in-hospital vaccination counselling, particularly during periods of vulnerability such as hospitalization. Consistent with previous research [20,21,22], our findings reinforce the role of hospitals as critical settings for reaching and empowering frail patients who often exhibit low adherence to preventive care in community settings. Indeed, hospital-based vaccination offers several advantages: timely identification of high-risk individuals, reduction in logistical and access barriers, assurance of safe administration, and strengthening of the physician–patient trust relationship. Among the recruited population, a significant factor influencing first-time vaccine uptake was the perception that vaccination would not interfere with the acute condition that led to hospitalization. This reassurance, provided directly by a healthcare worker, appears to play a key role in overcoming vaccine hesitancy, particularly among previously unvaccinated individuals.

Regarding the co-administration of the influenza and SARS-CoV-2 vaccines, almost 18% of participants in our study had a greater propensity to accept both vaccines during the same hospital stay. Although this rate may appear modest, it is consistent with other research findings, as those of a recent Canadian population-based survey, in which only 26.2% of respondents reported receiving both vaccines simultaneously during the 2022–2023 season [25]. Interestingly, uptake was even lower among high-risk groups, including older adults (23.9%) and individuals aged 18–64 with chronic conditions (25.4%), despite high levels of awareness (approximately 70%) regarding the feasibility of co-administration [25]. These data highlight that even in well-informed populations, vaccination compliance may be limited, depending on multiple factors, including psychological ones.

The HAPA questionnaire used in our study provided valuable insights into the psychological factors influencing vaccine co-administration. Specifically, patients who accepted co-administration were significantly more likely to report positive outcome expectations (aOR = 1.29, *p* = 0.02)—such as a reduced risk of hospitalization or pneumonia—and higher levels of self-efficacy (aOR = 1.48, *p* < 0.001), indicating that they felt well-informed and confident in their decision, even when facing potential opposition from family members or peers. These findings align with both U.S. and European literature, which consistently highlight the role of sociopsychological variables, alongside demographic, socioeconomic, and clinical factors, in shaping influenza vaccine adherence [26,27]. Notably, the social influence of healthcare professionals plays a central role, along with perceived vulnerability, perceived susceptibility to infection, perceived vaccine efficacy, and self-efficacy [27,28]. For instance, British studies have shown that individuals who accepted vaccination were more likely to base their health decisions on physicians’ recommendations [26]. Moreover, Opel et al. found that patients were more likely to refuse vaccination when physicians used a participatory communication style (e.g., “What do you want to do about vaccines?”) rather than a presumptive approach (e.g., “We should get vaccinated”) [29]. These findings underscore how the way physicians frame conversations about vaccination significantly influences patient behaviour.

It is therefore crucial to consider patients’ psychological profiles and the central role of the vaccination provider in addressing beliefs, concerns, and decision-making styles through a personalized and reassuring communication strategy. These sociopsychological dimensions offer valuable leverage points for developing targeted interventions. The influence of physicians’ communication style—independent of patients’ general trust in institutions—highlights the need to strengthen the physician–patient dialogue at the point of care. Evidence-based strategies such as personalized counseling and structured reminders, already shown to be effective in older populations, should be extended to younger, disadvantaged, or vaccine-hesitant groups [24]. Equally important is bridging the gap between perceived and actual risk by replacing generic messages with personalized risk communication based on individual clinical profiles [26]. For example, highlighting complications specifically relevant to young adults with diabetes may resonate more strongly than broadly targeted campaigns [26]. Such personalization enhances patients’ awareness of their vulnerability and increases their motivation to get vaccinated.

Considering this, integrating structured communication tools, and validated psychological models—such as HAPA—into hospital-based vaccination strategies can improve the quality of patient counseling and promote adherence, especially among frail or hesitant individuals. Despite this growing body of evidence, the Italian Ministry of Health’s guidelines for seasonal influenza vaccination do not yet explicitly endorse hospital-based active vaccine delivery, representing a missed opportunity to improve national coverage rates [21].

Another key finding was the strong association between prior influenza vaccination and current vaccine uptake. Individuals who had received the influenza vaccine in previous seasons were significantly more likely to accept vaccination during hospitalization. This is consistent with several studies identifying vaccination history as one of the strongest predictors of current vaccine acceptance [21,30,31,32,33]. This association reflects behavioural reinforcement, where past positive vaccination experiences foster the perception of vaccines as safe, effective, and routine. Such individuals usually develop a “pro-vaccine habit,” increasing their readiness to receive future vaccinations, particularly in the context of seasonal immunization [33]. Conversely, those without previous vaccination experience may encounter psychological or informational barriers rooted in hesitancy, lack of awareness, or mistrust [21,34]. These findings highlight the importance of emphasizing positive past experiences in public health communication and clinical counselling. Then, hospital interventions should aim to engage newly vaccinated individuals as well as support and strengthen adherence among those with a history of vaccination. By leveraging the predictive value of prior vaccine uptake, public health strategies can be more effectively tailored to individual profiles, particularly to support vulnerable populations at greater risk of severe influenza outcomes.

Older age (≥65 years) was also marginally associated with higher vaccine uptake and higher vaccine co-administration. This supports age as both a marker of increased risk perception and a key criterion in public health vaccination priorities. Interestingly, research conducted before the COVID-19 pandemic also shows a tendency for older adults to be more willing to accept seasonal influenza vaccination [35]. As aging is associated with a higher prevalence of chronic conditions, this group probably perceives vaccination as a necessary protective measure. Public health bodies—including the WHO and national authorities—have long prioritized adults aged 65+ for influenza vaccination due to their elevated morbidity and mortality risks [3,12]. Our results support these recommendations and reinforce the importance of targeted, proactive strategies for this demographic, especially during hospitalization.

Finally, a higher level of education was associated with a threefold increase in adherence to the simultaneous administration of influenza and SARS-CoV-2 vaccines (aOR = 4.21, *p* = 0.002). This finding may reflect greater health literacy and improved access to reliable health information. Moreover, our results are consistent with previous studies showing that higher educational attainment is linked to greater knowledge about vaccination and a more positive attitude toward receiving the influenza vaccine or vaccinations in general [21,36,37].

Although this was a multicentric study conducted in three hospitals, limitations such as potential selection bias and low acceptance of co-administration may restrict the generalizability of findings. Moreover, new efforts should include follow-up of recruited patients and a second survey in the following vaccination season to evaluate the long-term effects of both vaccine adherence and refusal. Future research should include digital health information-seeking behavior, physical activity level, and additional lifestyle and psychological factors—such as alcohol consumption, stress, and anxiety—as complementary indicators of health consciousness, as all these aspects may influence individuals’ attitudes and adherence to vaccination. Implementing longitudinal monitoring or repeating the study across subsequent seasons would allow for a more comprehensive evaluation of behavioural trends and the sustained impact of hospital-based interventions over time.

## 5. Conclusions

This study explored the impact of influenza vaccination coverage in implementing in-hospital vaccination offer. The reliability of this strategy, together with the standard vaccination offer, could allow reaching the recommended vaccination coverage, particularly among at-risk people. Implementing in-hospital vaccination service could also increase vaccine co-administration adherence. Moreover, education of hospital HCWs about vaccination and advising hospitalized patients should be a more effective strategy to provide tailored counselling about vaccination of hospitalized patients.

## Data Availability

Data will be available upon a motivated request to the corresponding author.

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
