# Peer review of "Effectiveness of an Active Offer of Influenza Vaccination to Hospitalized Frail Patients"

_vaccines, 2025, doi:10.3390/vaccines13111165_

Round 1

Reviewer 1 Report

Comments and Suggestions for Authors

The manuscript addresses a highly relevant and compelling topic. The presentation of the material is very good, and both the methodology and the reported results are articulated with exemplary clarity and precision.

The authors have thoroughly discussed the findings and have effectively addressed the core issues in the Discussion section. Their analysis is robust and well-supported.

In my  opinion, the scientific merit and clarity of presentation meet the journal's standards. I therefore recommend that the article be accepted for publication.

Author Response

Comments 1: The manuscript addresses a highly relevant and compelling topic. The presentation of the material is very good, and both the methodology and the reported results are articulated with exemplary clarity and precision.

The authors have thoroughly discussed the findings and have effectively addressed the core issues in the Discussion section. Their analysis is robust and well-supported.

In my  opinion, the scientific merit and clarity of presentation meet the journal's standards. I therefore recommend that the article be accepted for publication.

Response 1: We sincerely thank the reviewer for the positive and encouraging comments.

4. Response to Comments on the Quality of English Language

Point 1: The English is fine and does not require any improvement.

Response 1: Not applicable

Reviewer 2 Report

Comments and Suggestions for Authors

An interesting study...I do have some observations:

1) Was knowing someone who had died or had serious complications from influenza, recorded. I think that would have been interesting to look at 

2) I think a factor that needs to be addressed is level of computer use for researching vaccinations or illnesses. Many older adults do this

3) Physical activity levels? You have questions on nutrition, but what about physical activity. You may find those that are physically active (e.g., walking) are more likely to be health conscious and then more likely to be vaccinated

4) Was alcohol levels or stress/anxiety measured?

Author Response

3. Point-by-point response to Comments and Suggestions for Authors

Comments 1: Was knowing someone who had died or had serious complications from influenza, recorded. I think that would have been interesting to look at.

Response 1: We sincerely thank the reviewer for this insightful comment. We fully agree that knowing someone who experienced severe influenza complications or death could represent an important determinant of vaccination behavior, as personal experiences may strongly influence individual risk perception and preventive attitudes. This variable could indeed be valuable to include in future studies on this topic.

Although our questionnaire did not directly assess personal acquaintance with individuals who had serious influenza-related outcomes, the survey did capture participants’ perceived risk and outcome expectancies through the constructs of the Health Action Process Approach (HAPA) model. Specifically, respondents were asked to express their agreement with statements such as: “diseases such as influenza, COVID-19, pneumonia, or shingles could significantly worsen your health”, “influenza, COVID-19, pneumonia, or shingles could lead to a new hospitalization”, “receiving two vaccines at the same time may reduce your risk of hospitalization due to disease-related complications”. These items indirectly reflect how individuals appraise the severity and personal relevance of infectious diseases, which can be influenced by prior experiences or awareness of severe cases within one’s social network.

Comments 2: I think a factor that needs to be addressed is level of computer use for researching vaccinations or illnesses. Many older adults do this

Response 2: We appreciate the reviewer’s thoughtful comment. We agree that the level of computer use and the habit of seeking health-related information online represent important aspects of digital health literacy, which can substantially influence vaccination attitudes and decision-making, particularly among older adults. Individuals who actively search for medical information on the internet may develop stronger opinions—either positive or negative—depending on the reliability of the sources consulted.

Although our current study did not include a specific measure of online health information–seeking behavior, this factor is highly relevant and will be considered in future research. In the present work, related cognitive dimensions were indirectly captured through the Health Action Process Approach (HAPA) constructs, which assessed perceived risk, expected outcomes, and self-efficacy regarding vaccination. These psychological determinants are often shaped by exposure to information, including digital sources.

We have added the following text to the revised manuscript:

Lines 400-404: “Future research should include digital health information–seeking behavior, physical activity level, and additional lifestyle and psychological factors—such as alcohol consumption, stress, and anxiety—as complementary indicators of health consciousness, as all these aspects may influence individuals’ attitudes and adherence to vaccination.”

Comments 3: Physical activity levels? You have questions on nutrition, but what about physical activity. You may find those that are physically active (e.g., walking) are more likely to be health conscious and then more likely to be vaccinated

Response 3: We appreciate the reviewer's insightful observation. Physical activity was not included in our questionnaire; however, we agree that it represents a relevant indicator of general health awareness, which may be related to vaccination adherence. We have added the following text to the revised manuscript:

Lines 400-404: “Future research should include digital health information–seeking behavior, physical activity level, and additional lifestyle and psychological factors—such as alcohol consumption, stress, and anxiety—as complementary indicators of health consciousness, as all these aspects may influence individuals’ attitudes and adherence to vaccination.”

Comments 4: Was alcohol levels or stress/anxiety measured?

Response 4: We thank the reviewer for highlighting these additional health-related factors. Measures of alcohol consumption and psychological well-being were not part of the survey instruments. We have added the following text to the revised manuscript:

Lines 400-404: “Future research should include digital health information–seeking behavior, physical activity level, and additional lifestyle and psychological factors—such as alcohol consumption, stress, and anxiety—as complementary indicators of health consciousness, as all these aspects may influence individuals’ attitudes and adherence to vaccination.”

4. Response to Comments on the Quality of English Language

Point 1: The English is fine and does not require any improvement.

Response 1: Not applicable

Reviewer 3 Report

Comments and Suggestions for Authors

This is a paper on an important subject: influenza vaccination among older or at-risk individuals. The main reported finding is of clear health-care interest: personal recommendations by doctors can considerably boost vaccination uptake. The paper is clearly written. 

However, in my view there are two main problems with the research design. First, it is not clear how the people who got personal advice from hospital healthcare workers were selected. It seems that this intervention was not experimentally controlled, but registered afterwards. If so, there is clear possibility of selection bias: such advice may have been given more to patients who were at high risk, or who seemed more receptive to such advice. (Also, memory effects may play a role: persons who followed up on the advice may be more likely to remember or report it.) This is neither acknowledged nor discussed. 

The second problem is that it is not clear how older people who were already vaccinated before entering the hospital were treated. As the study period was between November 2023 and February 2024, this is likely to have often happened. Most likely and properly, they were considered not eligible. This would also induce a selection bias, which should be acknowledged and discussed. 

Author Response

3. Point-by-point response to Comments and Suggestions for Authors

Comments 1: This is a paper on an important subject: influenza vaccination among older or at-risk individuals. The main reported finding is of clear health-care interest: personal recommendations by doctors can considerably boost vaccination uptake. The paper is clearly written.

However, in my view there are two main problems with the research design. First, it is not clear how the people who got personal advice from hospital healthcare workers were selected. It seems that this intervention was not experimentally controlled, but registered afterwards. If so, there is clear possibility of selection bias: such advice may have been given more to patients who were at high risk, or who seemed more receptive to such advice. (Also, memory effects may play a role: persons who followed up on the advice may be more likely to remember or report it.) This is neither acknowledged nor discussed.

The second problem is that it is not clear how older people who were already vaccinated before entering the hospital were treated. As the study period was between November 2023 and February 2024, this is likely to have often happened. Most likely and properly, they were considered not eligible. This would also induce a selection bias, which should be acknowledged and discussed.

Response 1: We sincerely appreciate the reviewer's comments.
We clarify that the influenza vaccination offering was systematically delivered to all eligible inpatients (i.e., adults aged ≥60 years or with chronic conditions) at hospital discharge, irrespective of their health status, perceived risk, or individual attitudes. Furthermore, Italian guidelines recommend that healthcare workers advise influenza vaccination to all at-risk patients and the question of the manuscript was addressed to explore if healthcare workers provided the advice, not the quality of the advice related to the severity of health conditions. Therefore, we can assume that the selection process not have a greater role in determining which patients received advice or an offer of vaccination. Moreover, during hospital stay all hospitalized individuals meeting the eligibility criteria were informed about the availability, risks, and benefits of influenza vaccination by the same trained healthcare workers.

To clarify better, the questionnaire was administered before hospital discharge. Consequently, information on whether the individual received a personal recommendation from a hospital healthcare worker and whether they accepted vaccination was recorded directly, minimizing the risk of recall bias.

As for individuals already vaccinated before hospital admission, this information was collected by verifying clinical records at discharge. These individuals were classified as “already vaccinated” in the analysis but were still included in the study population to accurately describe overall vaccination coverage and behaviors. Indeed, influenza vaccination in Sicily was allowed by guidelines both outside and inside hospitals, and the possibility of being vaccinated can also depend on accessibility to vaccination service due to distance or health conditions.

Because the vaccination offer was extended to all hospitalized eligible individuals, regardless of previous vaccination status, there was no exclusion or differential selection that could have introduced systematic bias. Nevertheless, we have clarified this aspect in the Methods section as follows:

Lines 110-114: “Influenza vaccination was systematically offered to all eligible hospitalized patients (aged ≥60 years or with chronic conditions) before hospital discharge, independently of their level of frailty or personal attitudes. Information on previous influenza vaccination and receipt of in-hospital recommendations was collected through direct interviews and medical record verification, minimizing the risk of selection or recall bias.”

Once again, we thank the reviewer for providing the opportunity to clarify these methodological details, which we believe strengthen the transparency and reproducibility of our study.

4. Response to Comments on the Quality of English Language

Point 1: The English is fine and does not require any improvement.

Response 1: Not applicable

Reviewer 4 Report

Comments and Suggestions for Authors

This study evaluated the impact of actively offering influenza vaccination to hospitalized frail patients (≥60 years or with chronic conditions) in three Sicilian hospitals. Of 418 participants, 58.4% accepted the vaccine. Key factors increasing uptake were older age, recommendations from primary care or hospital healthcare workers, and previous influenza vaccination. Only 17.9% showed high propensity for vaccine co-administration, which was associated with higher education and positive psychological factors like self-efficacy. The findings demonstrate that proactive, in-hospital vaccination offers are a highly effective strategy for increasing coverage among high-risk populations in regions with low vaccination rates.

Here are my concerns regarding this study:

- Is there a sample size inconsistency in the Abstract and Results section regarding the number of patients who received the influenza vaccine? In the Abstract, it states, "A total of 418 hospitalized patients were enrolled in the study, of whom 58.4% (n = 224) received the influenza vaccine..." However, in the Results (Table 2 and text), it states "Accepted administration of Flu Vaccine (n = 244, 58.4%)". Please clarify, as all subsequent statistics and conclusions based on the vaccinated group's size might be called into question until this error is corrected.

- This study is focused on vaccination uptake (the process of getting vaccinated), not on the health outcomes (like infection, hospitalization, or death) that result from vaccination. Does this study measure or demonstrate a benefit of actively offering influenza vaccination in terms of reduced infection rates? The current study relies on the well-established, pre-existing evidence from other studies that the influenza vaccine itself is effective at reducing infection and severe outcomes. But, how to best get that effective vaccine into the arms of those who need it most (vulnerable population, including the hospitalized frail patients).

- How did you address the "Not sure" responses for variables like "Previous influenza vaccination," where the category was included in the univariable analysis but dropped in the multivariable model without explanation? Could this approach introduce bias if the "Not sure" group is systematically different?

- Could you clarify the approach used for variable selection in the multivariable models? The manuscript mentions that "all variables that were significantly associated" in the univariable analysis were included, but this method might exclude important confounding variables that were not statistically significant in the initial screening. How did you account for this potential limitation?

Author Response

3. Point-by-point response to Comments and Suggestions for Authors

Comments 1: Is there a sample size inconsistency in the Abstract and Results section regarding the number of patients who received the influenza vaccine? In the Abstract, it states, "A total of 418 hospitalized patients were enrolled in the study, of whom 58.4% (n = 224) received the influenza vaccine..." However, in the Results (Table 2 and text), it states "Accepted administration of Flu Vaccine (n = 244, 58.4%)". Please clarify, as all subsequent statistics and conclusions based on the vaccinated group's size might be called into question until this error is corrected.

Response 1: We thank the reviewer for noticing this inconsistency. The reviewer is absolutely right — the discrepancy is due to a typographical error in the Abstract. The correct number of participants who received the influenza vaccine is 244 (58.4%), as correctly reported in the Results section and in Table 2.

All statistical analyses and conclusions presented in the manuscript were performed using the correct value (n = 244). We have now corrected the Abstract accordingly to ensure consistency throughout the manuscript.

Line 24-25: “A total of 418 hospitalized patients were enrolled in the study, of whom 58.4% (n = 244)….

Comments 2: How did you address the "Not sure" responses for variables like "Previous influenza vaccination," where the category was included in the univariable analysis but dropped in the multivariable model without explanation? Could this approach introduce bias if the "Not sure" group is systematically different?

Response 2: We thank the reviewer for this important observation. As suggested, we repeated the multivariate analysis, including the "Not sure" category, to assess whether its exclusion might have influenced the results. The results remained essentially unchanged showing the consistency of results. Accordingly, we modified the Results and Discussion sections as follows:

Lines 224-229: “The multivariable logistic regression model for influenza vaccination acceptance showed that having received advice from hospital healthcare workers (aOR = 10.6; 95% CI: 5.33–20.9; p < 0.001) and previous influenza vaccination (aOR = 18.1; 95% CI: 7.98–41.1; p < 0.001) were both significantly associated with current vaccine uptake.”.

Lines 282-284:“Multivariate analysis confirmed the pivotal role of hospital-based interventions. Receiving a vaccination recommendation from hospital staff was associated with a more than tenfold increase in the likelihood of influenza vaccine uptake (aOR = 10.6; 95% CI: 5.33–20.9; p < 0.001), consistent with findings from other investigations on inpatient vaccination adherence [20] [23]”.

Comments 3: Could you clarify the approach used for variable selection in the multivariable models? The manuscript mentions that "all variables that were significantly associated" in the univariable analysis were included, but this method might exclude important confounding variables that were not statistically significant in the initial screening. How did you account for this potential limitation?

Response 3: We appreciate the comments, but considering the number of recruited patients and that including all variables can confound results, we chose to include in the multivariable model only the significant variables at univariate analysis to have only the main effects in the multivariable model, other than age and sex, which in the literature are reported to determine influenza vaccine acceptance.

4. Response to Comments on the Quality of English Language

Point 1: The English is fine and does not require any improvement.

Response 1: Not applicable

Round 2

Reviewer 4 Report

Comments and Suggestions for Authors

The author has thoroughly addressed all the concerns raised and made the necessary revisions to improve the manuscript. The current version reflects a clear, well-structured, and comprehensive response to feedback, meeting all required standards. As such, the manuscript is now in good condition and ready for publication.

Author Response

Thank you for your reply